# Factors associated with and socioeconomic inequalities in breast and cervical cancer screening among women aged 15–64 years in Botswana

**Mpho Keetile**[1]*, **Kagiso Ndlovu**[2], **Gobopamang Letamo**[1], **Mpho Disang**[3], **Sanni Yaya**[4], **Kannan Navaneetham**[1]

1 Department of Population Studies, University of Botswana, Gaborone, Botswana, 2 Department of Computer Science, University Botswana, Gaborone, Botswana, 3 The University of Edinburgh, Edinburgh, United Kingdom, 4 International Development and Global Studies, Faculty of Social Sciences, University of Ottawa, Ottawa, Canada

* mphokeet@yahoo.com

## Abstract

### Background

The most commonly diagnosed cancers among women are breast and cervical cancers, with cervical cancer being a relatively bigger problem in low and middle income countries (LMICs) than breast cancer.

### Methods

The main aim of this study was to asses factors associated with and socioeconomic inequalities in breast and cervical cancer screening among women aged 15–64 years in Botswana. This study is part of the broad study on Chronic Non-Communicable Diseases in Botswana conducted (NCD survey) in 2016. The NCD survey was conducted across 3 cities and towns, 15 urban villages and 15 rural areas of Botswana. The survey collected information on several NCDs and risk factors including cervical and breast cancer screening. The survey adopted a multistage sampling design and a sample of 1178 participants (males and females) aged 15 years and above was selected in both urban and rural areas of Botswana. For this study, a sub-sample of 813 women aged 15–64 years was selected and included in the analysis. The inequality analysis was conducted using decomposition analysis using ADePT software version 6. Logistic regression models were used to show the association between socioeconomic variables and cervical and breast cancer screening using SPSS version 25. All comparisons were considered statistically significant at 5%.

### Results

Overall, 6% and 62% of women reported that they were screened for breast and cervical cancer, respectively. Women in the poorest (AOR = 0.16, 95% CI = 0.06–0.45) and poorer (AOR = 0.37, 95% CI = 0.14–0.96) wealth quintiles were less likely to report cervical cancer

**Data Availability Statement:** The data underlying the results presented in the study are available from the Department of Population Studies Data Repository, University of Botswana.Contact; Dr

Enock Ngome (Head of Department) Email: ngome@ub.ac.bw Telephone:+267 3552710

**Funding:** This study was funded by the Office of Research and Development (ORD) at the University of Botswana and any request for data access may be sent to the ORD.

**Competing interests:** The authors have declared that no competing interests exist.

screening compared to women in the richest wealth quintile. Similarly, for breast cancer, the odds of screening were found to be low among women in the poorest (AOR = 0.39, 95% CI = 0.06–0.68) and the poorer (AOR = 0.45, 95% CI = 0.13–0.81)) wealth quintiles. Concentration indices (CI) showed that cervical (CI = 0.2443) and breast cancer (CI = 0.3975) screening were more concentrated among women with high SES than women with low SES. Wealth status was observed to be the leading contributor to socioeconomic inequality observed for both cervical and breast cancer screening.

## Conclusions

Findings in this study indicate the need for concerted efforts to address the health care needs of the poor in order to reduce cervical and breast cancer screening inequalities.

## Introduction

The most commonly diagnosed cancers among women globally are breast and cervical cancers [1], with cervical cancer being a relatively bigger problem in low and middle income countries (LMICs) than breast cancer [2]. It is estimated that globally, more than 2 million women are diagnosed with breast or cervical cancer each year [3]. Of this proportion, approximately 85% of women diagnosed and 88% of women who die from cervical cancer are from LMICs [3]. Several factors are often at play when it comes to prevalence of cervical or breast cancer screening. Literature indicates that whether a woman develops cancers, how early it manifests, whether she has access to safe and affordable diagnostic and treatment services is a function of her place of residence, in which country, in areas remote from health care services, and how she lives—poor or otherwise socially disenfranchised [1].

While cervical cancer risk factors are modifiable, some evidence suggests that breast cancer risk factors are not modifiable or are hard to control at the population level, as a result early detection, as well as proper treatment, is indispensable for improving the disease prognosis [4]. In many countries, particularly LMICs the Papanicolaou (PAP) smear test is the most common strategy employed to reduce cervical cancer incidence and mortality [5]. The three screening methods for breast cancer include breast self-examination (BSE), physical examination of the breasts by a physician or qualified health workers or clinical breast examination (CBE) and mammography (HICs) [5]. Cervical and breast cancers are largely preventable diseases in most HICs due to screening, early detection, and treatment, while in LMICs many women still lack information about these diseases [6]. Similarly there is lack of information about their risk factors, access to screening and early treatment, and oncology services for the treatment of advanced diseases [7].

The Southern Africa region ranked third among regions in cervical cancer deaths in 2012, with an age-standardized rate (ASR) of 17.9 women per 100,000 compared to a worldwide ASR of 6.8 per 100,000 [8]. Since early detection and treatment of cervical and breast cancers play a significant role in determining outcomes for women with cervical or breast cancer, considerable attention has been paid by health professionals and researchers to promote women's utilization of prevention services in the region. For instance, several studies conducted in South Africa, Zimbabwe and Zambia has shown that there has been a scale-up of screening [9]. Meanwhile several factors have been linked to cancer screening. They include being age, education, insurance, wealth status, culture, income, misconceptions about cervical or breast

cancer, and perceived vulnerability to the disease [10–12]. Some studies in LMICs have linked low levels of knowledge and uptake of cervical and breast cancer screening to low socio-economic status (SES) [11, 13, 14]. Consequently, inequalities in the use of breast and cervical cancer screening services due to SES have been detected in some settings [11, 12], with more deprived women less likely to be screened. For instance, in Europe a study comparing cervical and breast cancer screening inequalities by educational level found that inequality is not a generalized phenomenon [14].

Botswana like many other LMICS is faced with the increasing cases of cervical and breast cancers. The last two decades have seen remarkable increases in cervical and breast cancer cases. As a result the government came with a response plan which includes cervical cancer screening in the form of cytology [14]. Even at that, access to early screening and early treatment, and oncology services for the treatment of advanced cancer is still a challenge, especially for disadvantaged women. Furthermore, cancer screening programs have had a limited impact on cervical cancer incidence and mortality due to challenges with follow-up, as well as pathological and treatment capacity in a setting where screening results are positive for a high proportion of women [14]. Moreover, while the Ministry of Health and Wellness (MOHW) has recognized the need to substantially improve and support cancer screening in the rising burden of cancers in general, cervical cancers and breast screening inequalities can be reduced.

It has been reported that in Botswana most women only seek medical help in the late stage of the disease [12]. This has been attributed to inadequate knowledge of breast and cervical cancer examination. Few studies have attempted to examine the extent of socioeconomic inequalities in breast and cervical cancer screening among adult women in the context of a universal primary health care setting. Understanding the extent of and examining socioeconomic inequalities in breast and cervical cancer screening among women is vital for the formulation of intervention strategies to reduce morbidity and mortality due to cancer among the disadvantaged women.

## Materials and methods

### Ethical statement

The study proposal along with the survey instruments were submitted to and approved by the Institutional Review Board of the University of Botswana (Ref #: UBR/RES/IRB/1583) for ethical clearance. Ethical clearance and research permit were also obtained from the Government of Botswana through the Ministry of Health and Wellness (Ref #: HPDME: 13/18/1 Vol. X (130). Informed written consent was sought from all eligible participants before the interviews. Privacy and confidentiality of the highest standard was maintained throughout the study by keeping the respondents anonymous.

### Setting

This study is part of the broad study on Chronic Non-Communicable Diseases in Botswana conducted (NCD survey) by Department of Population Studies in 2016. The NCD survey was conducted across 3 cities and towns, 15 urban villages and 15 rural areas of Botswana. The survey collected information on several other NCDs and NCD risk factors which were not covered in the WHO STEPS Botswana Surveys including cervical and breast cancer.

### Study design and sampling

The NCD survey used a mmultistage probability sampling design to collect data on various types of NCDs as stated in the ICD-10 classification of NCDs [15]. Four stages of sampling

were employed for data collection. First, the census districts were divided into rural and urban clusters. At the second stage, urban districts were divided into cities or towns and urban villages while rural clusters were maintained. Thirdly, a random selection of 3 cities and towns in the cities and towns strata, 15 urban villages from urban villages' strata and 15 rural villages' from the rural areas strata was made.

The final and fourth stage was the selection of enumeration areas using probability proportional to size sampling method for the different strata and localities. Then for every selected enumeration area (EA), 20 households were selected using systematic sampling method. This was done based on the guidelines used in most demographic health surveys (DHS) where 20–25 households (HHS) are selected from the primary sampling units (PSUs) [16]. The modified de facto type of enumeration was adopted for the NCD survey, whereby respondents above 15 years old were enumerated at the place where they were found at the time of survey using the interviewer (canvasser) method. At an EA, the coin was tossed to determine the cardinal point where the enumeration would start. The first household to be interviewed was determined using the day code. For instance, on the 29th March 2016 –the first household to be enumerated would be the 7th household from the farthest point of the EA. This code was arrived at by adding the digits 2 and 7. After this procedure, the listing of all eligible respondents aged 15 years and above was done. If there was more than one eligible participant in the selected household, one older person was selected to participate by a lottery method. If the eligible older person was absent during the first data collection visit, the interviewer arranged to return at another time to do the interview.

From an estimated initial sample size of 1280 respondents, a total of 1178 respondents aged 15 years and over who had successfully completed the NCD survey individual questionnaire were interviewed. For this study, a selection of women aged 15–64 years was made using SPSS version 27 data selection command. The inclusion criteria were such that all women who had successfully responded to questions on breast and cervical cancer screening were considered for analysis, while those who did not were excluded from the sample. The final sample size used for this study is 813 women aged 15–64 years.

## Measurement of variables

**Outcome variables.** *Cervical cancer screening*. This variable was computed from the question "The last time you had the pelvic examination; did you have a PAP smear test?" The resultant variable was such that women who did PAP smear were given a code 1 and 0 if otherwise.

*Breast cancer screening*. For this variable the question asked was 'when was the last time you had a mammography, if ever? A code of 1 was given to women who ever once did a mammography and 0 was given to those who never did mammography.

## Explanatory variables

Based on literature review [17], variables such as age, marital status, work status, residence, wealth status and education were considered as explanatory variables. Age was categorised into 15–24, 25–34, 35–49 years; marital status as married and not married; residence as urban and rural; and education as primary or lower, secondary and tertiary or higher. Wealth status was created from information on ownership of durable assets collected from respondents during the survey (e. g. ownership of car, refrigerator, and television), housing characteristics (e. g. material of dwelling floor and roof, main cooking fuel), access to basic services (e. g. electricity supply, source of drinking water, sanitation facilities) and ownership of livestock (e.g. cattle, goats, sheep, horses, chickens). The principal component analysis was then used to derive wealth quintiles, which have five categories from the 1st to the 5th quintile (poorest to richest).

## Data analysis

Multiple data analyses techniques were employed to assess socioeconomic inequalities in breast and cervical cancer screening. First univariate and bivariate analyses was undertaken to describe the sample and patterns of cervical and breast cancer screening. Second, logistic regression analysis was used to assess the association between socioeconomic variables and cervical and breast cancer screening using SPSS version 27. Results of logistic regression models were presented as adjusted odds ratios (AOR) together with their 95% confidence intervals.

Third, analysis of inequalities in cervical and breast cancer screening was done using ADePT software (version 6). The concentration curves and concentration indices were used to assess inequalities in cervical and breast cancer screening among women. The cumulative shares of the cervical and breast cancer screening variables were plotted using concentration curves against the cumulative share of the wealth status variable. In order to calculate the cumulative percentages, wealth status was ranked from lowest to highest quintile. If cervical or breast cancer screening was equally distributed among women, the curve would be observed running from the bottom left hand corner to the top right-hand corner (a 45˚ line) which is known as the line of equality. On the other hand, if the share of cervical or breast cancer screening was low among the women of low SES, the concentration curve would lie below the line of equality [18, 19]. The degree of inequality was assessed by the distance of the curve from the line of equality. The further the curve is from this line, the greater the degree of inequality. The first case of socioeconomic inequality is the case in which women with high SES have a positive value of concentration index., while the second case, where the curve is above the diagonal line, is known as socioeconomic inequality which disadvantages the women of lower SES and the value of the concentration index is negative [20].

The value of the cervical and breast cancer screening assigned to each woman was taken to be a function of the socioeconomic category to which the woman belongs. The value of the concentration index ranges between − 1 to + 1. The index is 0 if there is no socioeconomic related inequality. The achievement index was also used with the concentration index to reflect the average level of cervical and breast cancer screening and the inequality in cervical and breast cancer screening between the low SES women and the high SES women. It is the weighted average of cervical and breast cancer screening of the women in the sample, in which higher weights are, attached to low SES women than to high SES women [18]. The larger value of the index is considered as higher health dis-achievement to one group of women than the other group.

# Results

## Sample description

From a total sample of 813, the highest proportion in the sample constituted women in ages 25–34 years (27.3%) and <24 years, respectively (Table 1). Moreover, the highest proportion of women in the sample were urban village residents (47%), had secondary education (43%), were unemployed (44.8%), never-married (70.2%) and were in the poorer wealth quintile (22.3%). Overall, 62% and 6% of women reported that they were screened for cervical and breast cancer, respectively.

## Prevalence of cervical and breast cancer screening

Table 2 shows the prevalence of cervical and breast cancer screening among women in the sample. The proportion of women who did cervical cancer screening increased with age until 64 years. For instance, 31% of women aged <24 years were screened for cervical cancer,

**Table 1. Sample description.**

| Variable | N (813) | % |
|---|---|---|
| **Age** | | |
| <24 | 190 | 23.4 |
| 25–34 | 222 | 27.3 |
| 35–44 | 156 | 19.2 |
| 45–54 | 124 | 15.3 |
| 55–64 | 75 | 9.2 |
| >65 | 44 | 5.5 |
| **Place of residence** | | |
| Cities and towns | 208 | 25.6 |
| Urban villages | 382 | 47.0 |
| Rural villages | 223 | 27.3 |
| **Education level** | | |
| Primary or less | 335 | 41.2 |
| Secondary | 350 | 43.0 |
| Tertiary or higher | 128 | 15.8 |
| **Work status** | | |
| Public sector | 76 | 9.3 |
| Private sector | 98 | 12.1 |
| Self-employed | 69 | 8.5 |
| Un-employed | 364 | 44.8 |
| Home-maker/student | 153 | 18.8 |
| Retired/other | 53 | 6.5 |
| **Marital status** | | |
| Never-married | 571 | 70.2 |
| Currently married | 149 | 18.3 |
| Formerly married | 93 | 11.5 |
| **Wealth status** | | |
| Poorest | 179 | 22 |
| Poorer | 181 | 22.3 |
| Middle | 166 | 20.4 |
| Richer | 152 | 18.7 |
| Richest | 135 | 16.6 |
| Breast cancer screening | | |
| Yes | 49 | 6.0 |
| No | 764 | 94.0 |
| Cervical cancer screening | | |
| Yes | 504 | 62.0 |
| No | 309 | 38.0 |

compared to 71% among women in ages 55–64 years. Cervical cancer screening was also noted to be significantly high among public sector employees (77.2%), currently married (75.3%) and richest women (79.5%). All the comparisons were statistical significant at 5% level.

## Correlates of breast and cervical cancer screening

Table 3 shows the adjusted odd ratios for the association between cervical and breast cancer screening and socioeconomic characteristics of sampled women. For both cervical and breast

**Table 2. Prevalence of cervical and breast cancer screening among women aged 10–64 years in Botswana, 2016.**

| Variable | Cervical cancer screening | | Breast cancer screening | |
|---|---|---|---|---|
| | n (%) | p-value | n (%) | p-value |
| **Age** | | 0.00 | | 0.03 |
| <24 | 21 (31.3) | | 5 (4.1) | |
| 25–34 | 80 (69.6) | | 10 (6.9) | |
| 35–44 | 63 (71.6) | | 6 (5.6) | |
| 45–54 | 55 (78.6) | | 6 (7.9) | |
| 55–64 | 22 (71.0) | | 4 (8.7) | |
| >65 | 5 (29.4) | | 2 (6.7) | |
| **Place of residence** | | 0.51 | | 0.88 |
| Cities and towns | 78 (65.0) | | 11 (6.4) | |
| Urban villages | 128 (62.7) | | 15 (5.4) | |
| Rural villages | 60 (57.7) | | 11 (6.4) | |
| **Education level** | | 0.06 | | 0.10 |
| Primary or less | 89 (58.6) | | 10 (4.0) | |
| Secondary | 113 (61.4) | | 16 (6.2) | |
| Tertiary or higher | 61 (73.5) | | 10 (10.0) | |
| **Work status** | | 0.00 | | 0.03 |
| Public sector | 44 (77.2) | | 6 (12.8) | |
| Private sector | 40 (65.6) | | 9 (12.2) | |
| Self-employed | 26 (72.2) | | 1 (1.7) | |
| Un-employed | 107 (62.6) | | 14 (5.1) | |
| Home-maker/student | 29 (39.7) | | 5 (3.9) | |
| Retired/other | 20 (71.4) | | 2 (5.9) | |
| **Marital status** | | 0.16 | | 0.99 |
| Never-married | 174 (59.0) | | 26 (6.1) | |
| Currently married | 67 (75.3) | | 7 (5.8) | |
| Formerly married | 25 (56.8) | | 4 (5.8) | |
| **Wealth status** | | 0.00 | | 0.02 |
| Poorest | 39 (43.8) | | 3 (2.2) | |
| Poorer | 48 (57.8) | | 8 (5.9) | |
| Middle | 62 (70.5) | | 10 (8.1) | |
| Richer | 55 (61.1) | | 7 (5.6) | |
| Richest | 62 (79.5) | | 9 (8.9) | |

cancer screening the odds of being screened significantly increased with age until ages 45–64 years, but declined thereafter. For instance, there were no significant variations in cervical cancer screening for women in ages less than 24 years, and the odds of screening for cervical cancer were 10 times (AOR = 10.1, 95% CI = 2.54–40.6) higher among individuals aged 45–54 years compared to women aged 65 years and above. Similarly the odds of breast cancer screening increased with age and were significantly high among women aged 45–54 years (AOR = 2.61, 95% CI = 1.23–3.64) compared to women aged 65 years and above.

The odds of cervical cancer screening were significantly low among students and home makers (AOR = 0.21, 95% CI = 0.06–0.75) compared to retired women. Meanwhile, the odds of breast cancer screening were significantly higher among public sector employees (AOR = 1.49, 95% CI = 1.12–5.23) than retired women. Women in the poorest (AOR = 0.16, 95% CI = 0.06–0.45) and poorer (AOR = 0.37, 95% CI = 0.14–0.96) wealth quintiles were less likely to report cervical cancer screening compared to women in the richest wealth quintile.

**Table 3. The adjusted odds ratios for the association between cervical and breast cancer screening and socioeconomic characteristics of sampled women, 2016.**

| | Cervical cancer screening | | Breast cancer screening | |
|---|---|---|---|---|
| **Variable** | **AOR** | **95% CI** | **AOR** | **95% CI** |
| Age | | | | |
| <24 | 1.03 | (0.22–4.81) | 0.22 | (0.02–2.06) |
| 25–34 | 5.62*** | (1.28–24.6) | 1.23*** | (1.06–3.11) |
| 35–44 | 8.45*** | (2.04–35.0) | 1.43*** | (1.13–3.24) |
| 45–54 | 10.1*** | (2.54–40.6) | 2.61*** | (1.23–3.64) |
| 55–64 | 5.61*** | (1.32–23.7) | 1.07 | (0.15–7.46) |
| >65 | 1.00 | | 1.00 | |
| Place of residence | | | | |
| Cities and towns | 1.00 | | 1.00 | |
| Urban villages | 0.91 | (0.49–1.68) | 0.63 | (0.24–1.67) |
| Rural villages | 0.68 | (0.33–1.41) | 1.32 | (0.44–3.96) |
| Education level | | | | |
| Primary or less | 0.81 | (0.32–2.02) | 0.25 | (0.06–1.08) |
| Secondary | 1.06 | (0.50–2.21) | 0.71 | (0.24–2.09) |
| Tertiary or higher | 1.00 | | 1.00 | |
| Work status | | | | |
| Public sector | 0.47 | (0.12–1.78) | 1.49*** | (1.12–5.23) |
| Private sector | 0.28 | (0.08–1.01) | 2.69 | (0.47–15.4) |
| Self-employed | 0.36 | (0.09–1.42) | 0.3 | (0.03–3.67) |
| Un-employed | 0.42 | (0.13–1.32) | 1.09 | (0.21–5.62) |
| Home-maker/student | 0.21*** | (0.06–0.75) | 0.68 | (0.10–4.56) |
| Retired/other | 1.00 | | 1.00 | |
| Marital status | | | | |
| Never-married | 0.65 | (0.21–1.97) | 0.97 | (0.20–4.65) |
| Currently married | 0.60 | (0.18–1.96) | 0.78 | (0.15–4.03) |
| Formerly married | 1.00 | | 1.00 | |
| Wealth status | | | | |
| Poorest | 0.16*** | (0.06–0.45) | 0.39*** | (0.06–0.68) |
| Poorer | 0.37*** | (0.14–0.96) | 0.45*** | (0.13–0.81) |
| Middle | 0.51 | (0.19–1.34) | 1.07 | (0.28–4.10) |
| Richer | 0.37 | (0.16–1.86) | 1.32 | (0.36–4.88) |
| Richest | 1.00 | | 1.00 | |

Note:

*** statistically significant at 5% level.

Similarly, for breast cancer, the odds of screening were found to be low among women in the poorest (AOR = 0.39, 95% CI = 0.06–0.68) and the poorer (AOR = 0.45, 95% CI = 0.13–0.81)) wealth quintiles. There were no statistically significant educational, residential and marital status variations in cervical and breast cancer screening among women.

## Inequalities in cervical and breast cancer screening

Table 4 below presents the results of the inequality measures of concentration indices (CI) and the standard achievement indices for cervical and breast cancer screening. In a scenario where the concentration index is high, the achievement index is expected to be low and vice versa

**Table 4. Concentration indices showing inequalities in cervical and breast cancer screening in Botswana, (2016).**

| Variable | Concentration Index (CI) | 95% Confidence intervals for CI | Standard achievement index |
|---|---|---|---|
| Cervical cancer screening | 0.2443 | (0.1003,0.4150) | 0.1224 |
| Breast cancer screening | 0.3975 | (0.1242–6.131) | 0.1442 |

[19]. The positive CI value of 0.2443 for cervical cancer screening shows that the inequality is skewed towards women of high SES and the corresponding standard achievement index is low. Similarly the positive CI value of 0.3975 indicates that breast cancer screening is concentrated among women of high SES. The positive CI value for breast cancer screening is also accompanied by low corresponding standard achievement index.

The concentration curves plotting the cumulative share of cervical and breast cancer screening variables against the proportional cumulative share of wealth index (SES) score of women are shown in Fig 1. The curve for both cervical and breast cancer screening lie below

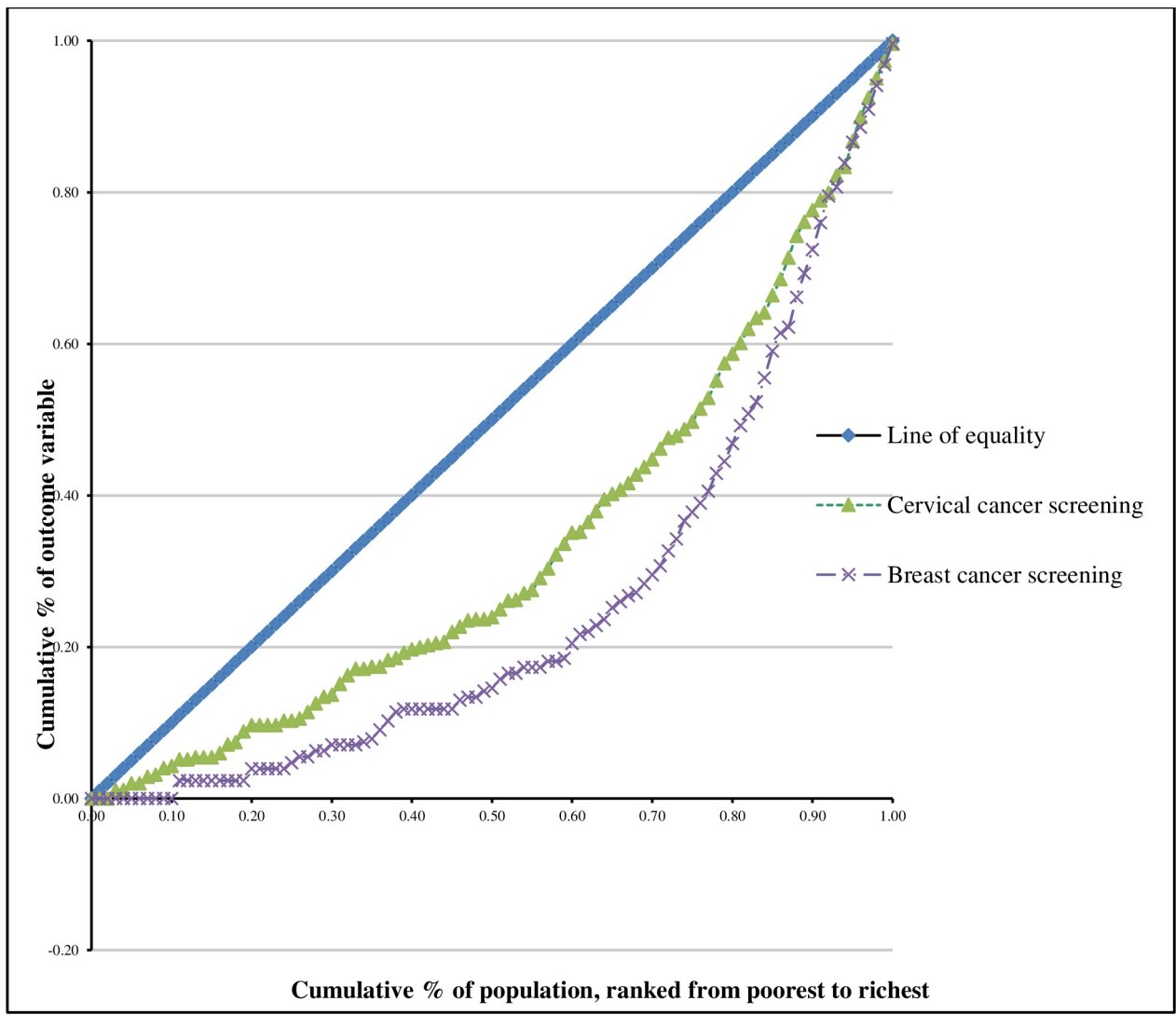

**Fig 1. Concentration curves for cervical and breast cancer screening-NCD survey, 2016.**

**Table 5. Decomposition of the concentration indices of the covariates for cervical and breast cancer screening variables.**

|  | Cervical cancer screening | Breast cancer screening |
| --- | --- | --- |
| **Covariates** | **coefficient** | **coefficient** |
| Residence | -0.2769 | -0.2352 |
| Age | -0.0528 | -0.0523 |
| Education | 0.1099 | 0.1031 |
| Marital status | -0.0088 | 0.1276 |
| Work status | 0.0256 | 0.0268 |
| **Control variables** | | |
| Wealth Index | 1.6420 | 3.0863 |

the line of equality which confirms that cervical and breast cancer screening was more concentrated among women of high SES. Meanwhile, the concentration curve for breast cancer screening is the furthest from the line of equality indicating that screening for breast cancer was slightly more concentrated among the women of high SES compared to cervical cancer screening. However, the inequality for both variables is significantly high.

## Decomposing inequalities in cervical and breast cancer screening

The decomposition of the concentration indices for cervical and breast cancer screening are shown in Table 5. These CIs explain why inequalities in cervical and breast cancer screening exist and what factors contribute to these observed inequalities. The results of decomposition show that variables such as residence, age and marital status had negative contribution to the inequalities for cervical and breast cancer screening. On the other hand, education, work status and wealth status itself dominated in the decomposition of the large concentration index for cervical and breast cancer screening. This was indicated by positive coefficients for these variables.

## Discussion

This study examined factors associated with and socioeconomic inequalities in breast and cervical cancer screening among women aged 15–64 years in Botswana. It emerged that majority of women had done cervical cancer screening than breast cancer screening. This is consistent with findings in many LMICs which have generally shown that cervical cancer is the second most common cancer in women in low- and middle-income countries (LMICs) [21–23]. Consequently screening rates for this type of cancer among women is comparatively high. This is mainly because in recent years, cervical cancer screening recommendations have relatively been updated and received more attention compared to mammography in Botswana.

The odds of being screened for cervical cancer were found to significantly increase with the age of women and were highest in ages 45–54 years, but declined thereafter. This finding is consistent with other previous studies which found that cervical cancer infection rates in women have generally been observed to peak in the years after sexual debut [24–26]. Consequently majority of women in these ages are more liable to screen for cervical cancer compared to women of younger ages. The American Cancer Society (ACS) recommends that women should undergo cervical cancer screening at age 25 years and undergo primary human papillomavirus (HPV) testing every 5 years through age 65 years (preferred); if primary HPV testing is not available, then women aged 25 to 65 years should be screened with co-testing (HPV testing in combination with cytology) every 5 years or cytology alone every 3 years [27].

Similarly the odds of breast cancer screening were found to increase with age of women but were highest in ages 45–54 years. This corroborates previous studies which have shown that women aged 40 years and above are at a greater risk of developing breast cancer [26–28]. Consequently they should under routine mammography, to enhance early detection of breast cancer. Although there are conflicting guidelines on the appropriate age for continuous breast cancer screening, the general consensus is that breast cancer screening intervals of 1–3 years for women aged 25–39 and annually for women aged 40 years and older are reasonable [28]. This offers a plausible explanation for high breast cancer screening rates among women aged 45–54 years in Botswana.

Consistent with previous studies [24, 26, 28–30], we found that women of low SES were found to be less likely to report both breast and cervical cancer screening compared to women of high SES. This is mainly because women from disadvantaged households are less likely to be knowledgeable and therefore unlikely to screen for cervical and breast cancer. This re-emphasizes the notion that those who have the financial means overcome barriers to accessing care compared to those who are poor. Unlike women of high SES, women of low SES do not have access to information and do not have access to insurance coverage. As a result providing health insurance to poor women may remove or reduce these financial barriers and guarantee a certain degree of equity in the use of cancer and breast screening services. Moreover, providing insurance to low SES women will give them the option to use private health facilities instead of few cervical and breast cancer screening public health facilities. Furthermore, the public health facilities are often congested and bookings for screening are delayed which is a major disadvantage to women of low SES who may require urgent screening. On the other hand there were no statistically significant educational, residential and marital status variations in cervical and breast cancer screening among women.

Inequalities in breast and cervical cancer screening among women were observed in Botswana despite the country-wide effort to improve the socioeconomic status and primary healthcare coverage of the population. The decomposition analysis shows that wealth status seems to be an important positive contributor to the concentration indices of the cervical and breast cancer screening outcomes. This means inequality in wealth makes cervical and breast cancer screening more predominant among richer individuals. Low uptake and disparities in cancer screenings could be attributed to the low level of awareness about the importance of early screening, which places a high proportion of low SES women under the risk of late detection.

## Study limitation

Certain limitations are worth mentioning. First, the cross-sectional nature of the data does not allow drawing causal inferences. Secondly, the NCD study sample was not designed to be representative of Botswana, a caution should be taken while generalizing these study findings.

## Supporting information

**S1 File.**
(DOCX)

## Author Contributions

**Conceptualization:** Mpho Keetile.

**Formal analysis:** Mpho Keetile.

**Methodology:** Gobopamang Letamo.

**Supervision:** Kannan Navaneetham.

**Validation:** Kagiso Ndlovu, Gobopamang Letamo, Mpho Disang, Sanni Yaya, Kannan Navaneetham.

**Writing – original draft:** Mpho Keetile, Kagiso Ndlovu, Gobopamang Letamo, Mpho Disang, Sanni Yaya, Kannan Navaneetham.

**Writing – review & editing:** Mpho Keetile, Kagiso Ndlovu, Gobopamang Letamo, Mpho Disang, Sanni Yaya, Kannan Navaneetham.

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
