## [Decision Letter · Decision Letter 0]

10 Feb 2021

PONE-D-20-39250

Factors associated with and socioeconomic inequalities in breast and cervical cancer screening among women aged 15-64 years in Botswana

PLOS ONE

Dear Dr. Keetile,

Thank you for submitting your manuscript to PLOS ONE. After careful consideration, we feel that it has merit but does not fully meet PLOS ONE’s publication criteria as it currently stands. Therefore, we invite you to submit a revised version of the manuscript that addresses the points raised during the review process.

An expert in the field handled your manuscript, and we are appreciative of their time and contributions. Although interest was found in your study, some major comments arose during review. Please address ALL of the reviewer's comments in your revised manuscript.

We look forward to receiving your revised manuscript.

Kind regards,

Frank T. Spradley

Academic Editor

PLOS ONE

2.We note that you have indicated that data from this study are available upon request. PLOS only allows data to be available upon request if there are legal or ethical restrictions on sharing data publicly. For information on unacceptable data access restrictions, please see http://journals.plos.org/plosone/s/data-availability#loc-unacceptable-data-access-restrictions.

3.Thank you for stating the following financial disclosure:

"NO"

5.Thank you for submitting the above manuscript to PLOS ONE. During our internal evaluation of the manuscript, we found significant text overlap between your submission and the following previously published works, some of which you are an author.

- https://doi.org/10.1186/s12889-019-7405-x

- https://doi.org/10.1016/j.socscimed.2017.12.013

- https://doi.org/10.7314/APJCP.2015.16.15.6697

Please revise the manuscript to rephrase the duplicated text, cite your sources, and provide details as to how the current manuscript advances on previous work. Please note that further consideration is dependent on the submission of a manuscript that addresses these concerns about the overlap in text with published work.

Reviewers' comments:

Reviewer's Responses to Questions

**Comments to the Author**

1. Is the manuscript technically sound, and do the data support the conclusions?

Reviewer #1: Partly

2. Has the statistical analysis been performed appropriately and rigorously? 

Reviewer #1: I Don't Know

3. Have the authors made all data underlying the findings in their manuscript fully available?

Reviewer #1: Yes

4. Is the manuscript presented in an intelligible fashion and written in standard English?

Reviewer #1: No

5. Review Comments to the Author

Reviewer #1: The data seems to have been made to fit a pre-determined conclusion i.e. that socioeconomic inequalities will exist in screening outcomes. This is intuitive and expected. The paper can be strengthened by thinking through the structural inequalities that exist underlying these differences in screening practices. i.e why do women with lower socioeconomic means have lower screening uptake? What can be causing this difference? What is the availability of cancer screening services? Could it be that there is no facility within a certain geographical radius? The structures underlying the distribution of inequality for the populations examined is not explored well.

In essence, the finding that these differences exist are no surprise since this topic has been studied repeatedly in many different countries. What can add to the literature is an examination of WHAT is driving these in the context of Botswana. Some clear next steps can then also be derived which are currently missing from the paper. The authors draw on other literature to cite a lack of awareness or lack of health insurance. But these can be applied more directly to the data for more substantive next steps.

Would also advise the author to keep consistent with use of language. Terms commonly used in the literature are social disadvantage, and how the authors then use this term should be clearly defined. The authors use different language such as poor and wealthy intermixed with disadvantaged.

In the introduction the authors mention the 'problem'. This problem needs to be explained more clearly and tied to the paper more directly.

The authors mention that Pap smears can reduce cervical cancer incidence, this is factually incorrect as Pap smears promote early detection.

Typo with mammography which is abbreviated as HIC in the manuscript.

6. PLOS authors have the option to publish the peer review history of their article (what does this mean?). If published, this will include your full peer review and any attached files.

Reviewer #1: No

---

## [Author Response · Author response to Decision Letter 0]

19 May 2021

Response to reviewer’s comments

Comment

Reviewer #1: The data seems to have been made to fit a pre-determined conclusion i.e. that socioeconomic inequalities will exist in screening outcomes. This is intuitive and expected. 

Response

Thank you for the comment. Kindly note that this study was derived from the secondary analysis of data for a study conducted by Department of Population Studies at University of Botswana, Faculty of Social Sciences, University of Botswana. The study proposal along with the necessary documents was submitted to and approved by the Institutional Review Board of the University of Botswana (Ref #: UBR/RES/IRB/1583) and the Ministry of Health and Wellness (Ref #: HPDME: 13/18/1 Vol. X (130)) therefore we did not make the data to fit any pre-determined conclusion. Although it may be true that inequalities may exist for the two outcomes, the extent of such inequalities is not known and in Botswana as far as we are concerned this is the first study to assess socioeconomic related inequalities in the context of a universal primary care setting. 

Comment

The paper can be strengthened by thinking through the structural inequalities that exist underlying these differences in screening practices. i.e why do women with lower socioeconomic means have lower screening uptake? What can be causing this difference? What is the availability of cancer screening services? Could it be that there is no facility within a certain geographical radius? The structures underlying the distribution of inequality for the populations examined are not explored well.

Response

Thank you so much. This is a valuable comment. We have indicated in the discussion the plausible explanation for observed inequalities. However kindly note that we cannot make causal inference from the cross-sectional data. The dataset did not have any structural variables, which could allow us to explore structural inequalities for breast and cervical cancer screening. However, we have discussed the plausible explanations for observed inequalities, including structural factors. This points the need for a broader study which can explore structural variables.

Comment

In essence, the findings that these differences exist are no surprise since this topic has been studied repeatedly in many different countries. What can add to the literature is an examination of WHAT is driving these in the context of Botswana. Some clear next steps can then also be derived which are currently missing from the paper. The authors draw on other literature to cite a lack of awareness or lack of health insurance. But these can be applied more directly to the data for more substantive next steps.

Response

Although we agree with the reviewers that socioeconomic inequalities for various outcomes are known in most countries, it is not the case in Botswana. Botswana is unique in the sense of universal health care coverage. Moreover, this is the first study to study inequalities, and as far as we are concerned it provides baseline evidence, especially on the extent of inequalities for the two outcomes. This is mainly because cervical and breast cancer screening is done for free in public health facilities. In the background we have also provided the context of why we think this study is credible, especially in Botswana

Comment

Would also advise the author to keep consistent with use of language. Terms commonly used in the literature are social disadvantage, and how the authors then use this term should be clearly defined. The authors use different language such as poor and wealthy intermixed with disadvantaged.

Response 

We have checked the language throughout the article and have corrected as per the reviewer’s comment. Kindly note that we have used the term socioeconomic inequality to refer to unjust socioeconomic differences in access and utilization of cervical and breast cancer screening. Previous studies have used the term socioeconomic inequality as used in the context of our study.

Comment

In the introduction the authors mention the 'problem'. This problem needs to be explained more clearly and tied to the paper more directly.

Response

We have made efforts to make the ‘problem’ clearer. We indicate quite clearly that given the context of Botswana inequalities in cervical and breast cancer screening are not expected. As a result this study serves to provide initial evidence on the extent of inequalities in screening.

Comment

The authors mention that Pap smears can reduce cervical cancer incidence, this is factually incorrect as Pap smears promote early detection.

Response

Thank you for the comment; we have duly corrected the statement

Comment

Typo with mammography which is abbreviated as HIC in the manuscript.

Response

Thank you we have made the observed correction

---

## [Decision Letter · Decision Letter 1]

14 Jun 2021

PONE-D-20-39250R1

Factors associated with and socioeconomic inequalities in breast and cervical cancer screening among women aged 15-64 years in Botswana

PLOS ONE

Dear Dr. Keetile,

Thank you for submitting your manuscript to PLOS ONE. After careful consideration, we feel that it has merit but does not fully meet PLOS ONE’s publication criteria as it currently stands. Therefore, we invite you to submit a revised version of the manuscript that addresses the points raised during the review process.

We look forward to receiving your revised manuscript.

Kind regards,

Frank T. Spradley

Academic Editor

PLOS ONE

Journal Requirements:

Reviewers' comments:

Reviewer's Responses to Questions

**Comments to the Author**

1. If the authors have adequately addressed your comments raised in a previous round of review and you feel that this manuscript is now acceptable for publication, you may indicate that here to bypass the “Comments to the Author” section, enter your conflict of interest statement in the “Confidential to Editor” section, and submit your "Accept" recommendation.

Reviewer #1: (No Response)

2. Is the manuscript technically sound, and do the data support the conclusions?

Reviewer #1: Yes

3. Has the statistical analysis been performed appropriately and rigorously? 

Reviewer #1: Yes

4. Have the authors made all data underlying the findings in their manuscript fully available?

Reviewer #1: Yes

5. Is the manuscript presented in an intelligible fashion and written in standard English?

Reviewer #1: Yes

6. Review Comments to the Author

Reviewer #1: Thank you for addressing the previous comments. The context of the study is now better understood. I still have some concerns around the use of language specifically. Given that the paper is describing socioeconomic inequalities, and that there is an abundance of critical literature that has contextualised this phenomenon, the authors must situate their paper in this body of work – and therefore most of my comments are related to how the authors describe their findings and position their work.

1. Labelling women as poor and wealthy detracts from the conditions leading to socioeconomic inequalities. It is better to use terms such as women living in marginalizing conditions, or women living in poverty. The term ‘poor’ women is used through out the paper and really should be avoided as it a non-critical label that otherizes.

2. In the introduction, authors mention that inequalities are inevitable – however, much contextual understanding now exists around cancer screening and the cancer care continuum which demonstrates that inequalities are indeed avoidable, unjust and unfair and that with the correct allocation of resources these inequalities can be reduced.

Finally, in the discussion the authors mention that health insurance can reduce financial barriers to screening – however, earlier in the paper it is mentioned that cancer screening in Botswana is a public health program provided at no cost to citizens. Please explain.

7. PLOS authors have the option to publish the peer review history of their article (what does this mean?). If published, this will include your full peer review and any attached files.

Reviewer #1: **Yes: **Ambreen Sayani

---

## [Author Response · Author response to Decision Letter 1]

16 Jul 2021

Response to reviewers comments-

Comment

Reviewer #1: Thank you for addressing the previous comments. The context of the study is now better understood. I still have some concerns around the use of language specifically. Given that the paper is describing socioeconomic inequalities, and that there is an abundance of critical literature that has contextualised this phenomenon, the authors must situate their paper in this body of work – and therefore most of my comments are related to how the authors describe their findings and position their work.

Response

Thank you so much for your invaluable comments. We appreciate that now you understand the context of our study. We have also made adjustment to our paper based on the local context as suggested by the reviewer.

Comment 

Labelling women as poor and wealthy detracts from the conditions leading to socioeconomic inequalities. It is better to use terms such as women living in marginalizing conditions, or women living in poverty. The term ‘poor’ women is used throughout the paper and really should be avoided as it a non-critical label that otherizes.

Response

Instead of using the word poor we have replaced it with low wealth/socioeconomic status as suggested. This is because wealth status variable was used a key dependent variable for measuring inequalities. Our understanding is that wealth status is a documented measure of socioeconomic status hence why we chose to use the word low socioeconomic status to refer to poor wealth status and vice versa.

Comment

 In the introduction, authors mention that inequalities are inevitable – however, much contextual understanding now exists around cancer screening and the cancer care continuum which demonstrates that inequalities are indeed avoidable, unjust and unfair and that with the correct allocation of resources these inequalities can be reduced.

Response

We have removed the statement that suggests that inequalities are inevitable but we emphasise that inequalities in cancer screening can be reduced.

Comment

Finally, in the discussion the authors mention that health insurance can reduce financial barriers to screening – however, earlier in the paper it is mentioned that cancer screening in Botswana is a public health program provided at no cost to citizens. Please explain.

Response

Although cervical and breast cancer screening is offered in public facilities , there are few specialised health facilities offering both breast and cervical cancer screening services, meaning that a high proportion of women especially in rural areas have limited access because the bookings for screening services can take as long as 6 months. Consequently this population group are disadvantaged compared to individuals who are on private medical insurance, who have the option to access screening services in private health facilities, which are often more apt. The delay experienced in screening women in public facilities due to limited human resource provides a challenge to low SES women. Our argument on providing medical insurance to poor women is derived from that background.

---

## [Editor Report · Decision Letter 2]

21 Jul 2021

Factors associated with and socioeconomic inequalities in breast and cervical cancer screening among women aged 15-64 years in Botswana

PONE-D-20-39250R2

Dear Dr. Keetile,

We’re pleased to inform you that your manuscript has been judged scientifically suitable for publication and will be formally accepted for publication once it meets all outstanding technical requirements.

Kind regards,

Frank T. Spradley

Academic Editor

PLOS ONE

---

## [Editor Report · Acceptance letter]

26 Jul 2021

PONE-D-20-39250R2 

Factors associated with and socioeconomic inequalities in breast and cervical cancer screening among women aged 15-64 years in Botswana 

Dear Dr. Keetile:

I'm pleased to inform you that your manuscript has been deemed suitable for publication in PLOS ONE. Congratulations! Your manuscript is now with our production department. 

Kind regards, 

on behalf of

Dr. Frank T. Spradley 

Academic Editor

PLOS ONE